# Radiosynthesis and Preclinical Evaluation of [^99m^Tc]Tc-Tigecycline Radiopharmaceutical to Diagnose Bacterial Infections

**DOI:** 10.3390/ph17101283

**Published:** 2024-09-27

**Authors:** Syeda Marab Saleem, Tania Jabbar, Muhammad Babar Imran, Asma Noureen, Tauqir A. Sherazi, Muhammad Shahzad Afzal, Hafiza Zahra Rab Nawaz, Mohamed Fawzy Ramadan, Abdullah M. Alkahtani, Meshari A. Alsuwat, Hassan Ali Almubarak, Maha Abdullah Momenah, Syed Ali Raza Naqvi

**Affiliations:** 1Department of Chemistry, Government College University Faisalabad, Faisalabad 38040, Pakistan; 2Punjab Institute of Nuclear Medicine, Faisalabad 38040, Pakistan; 3Department of Zoology, Ghazi University, Dera Ghazi Khan 03222, Pakistan; 4Department of Chemistry, COMSAT University Islamabad, Abbottabad Campus, Abbottabad 22060, Pakistan; 5Department of Clinical Nutrition, Faculty of Applied Medical Sciences, Umm Al-Qura University, Makkah 21955, Saudi Arabia; 6Department of Microbiology & Clinical Parasitology, College of Medicine, King Khalid University, Abha 61421, Saudi Arabia; 7Department of Clinical Laboratory Sciences, College of Applied Medical Sciences, Taif University, Taif 21944, Saudi Arabia; 8Assistant Professor Nuclear Medicine, Division of Radiology, Department of Medicine, College of Medicine and Surgery, King Khalid University, Abha 61421, Saudi Arabia; 9Department of Biology, College of Science, Princess Nourah bint Abdulrahman University, P.O. Box 84428, Riyadh 11671, Saudi Arabia

**Keywords:** antibiotics, tigecycline, radioisotope, nuclear medicine, radiopharmaceuticals, infection, *S. aureus*, *E. coli*, SPECT imaging

## Abstract

Background/Objectives: As a primary source of mortality and disability, bacterial infections continue to develop a severe threat to humanity. Nuclear medicine imaging (NMI) is known for its promising potential to diagnose deep-seated bacterial infections. This work aims to develop a new technetium-99m (^99m^Tc) labeled tigecycline radiopharmaceutical as an infection imaging agent. Methods: Reduced ^99m^Tc was used to make a coordinate complex with tigecycline at pH 7.7–7.9 at room temperature. Instantaneous thin-layer chromatography impregnated with silica gel (ITLC-SG) and ray detector equipped high-performance liquid chromatography (ray-HPLC) was performed to access the radiolabeling yield and radiochemical purity (RCP). Results: More than 91% labeling efficiency was achieved after 25 min of mild shaking of the reaction mixture. The radiolabeled complex was found intact up to 4 h in saline. *Staphylococcus aureus* (*S. aureus*) and *Escherichia coli* (*E. coli*) infection-induced rats were used to record the biodistribution of the radiopharmaceutical and its target specificity; 2 h’ post-injection biodistribution revealed a 2.39 ± 0.29 target/non-target (T/NT) ratio in the *E. coli* infection-induced animal model, while a 2.9 ± 0.31 T/NT value was recorded in the *S. aureus* bacterial infection-induced animal model. [^99m^Tc]Tc-tigecycline scintigraphy was performed in healthy rabbits using a single photon emission computed tomography (SPECT) camera. Scintigrams showed normal kidney perfusion and excretion into the bladder. Conclusion: In conclusion, the newly developed [^99m^Tc]Tc-tigecycline radiopharmaceutical could be considered to diagnose broad-spectrum bacterial infections.

## 1. Introduction

While the post-COVID-19 period is a rehabilitation period of certain psychological issues related to COVID-19 infection stress, it has also intensified the research to develop new protocols for the infection diagnosis and treatment. Different strategies are being used in clinical practice to diagnose viral, bacterial, and parasitic infections. These often depend on invasive protocols that do not reflect the true spatial heterogeneity of infection [1]. Instrumental modalities such as X-rays, ultrasonography (US), magnetic resonance imaging (MRI), and computed tomography (CT) scan, on the bases of morphological alteration in diseased tissues, provide anatomical, non-invasive, and accurate imaging of the infected tissues. However, morphological changes do not appear at early stages of disease [2]. Therefore, early-stage detection and understanding the root cause of lesion using these tools is impossible [3]. The NMI technique has been used since 1962 when ^99m^Tc was proposed as a useful radiotracer agent, and it has continuously gained momentum in diagnosing infections and metabolic disorders at stages prior to morphological alterations in infected tissues [4]. Therefore, continuous efforts have been made to introduce new radiopharmaceuticals to diagnose a variety of illnesses. Many heterocyclic organic compounds having good antibacterial activities and antibiotics of different classes and generations have been investigated for the development of infection imaging radiopharmaceuticals [5,6,7,8,9]. Primarily, [^67^Ga]citrate [10] and ^125^I/^111^In labeled nonspecific human polyclonal immunoglobulin (IgG) and liposomes were developed [11]. But, due to their nonspecific target accumulation behavior, later on specific radiotracers such as [^111^In]Oxine-labeled leukocytes [12], [^99m^Tc]Tc-HMPAO-labeled leukocytes [13], [^99m^Tc]Tc-chemotactic peptides [14], [^18^F]FDG [15], [^99m^Tc/^67^Ga]antimicrobial peptides [16], and [^99m^Tc]Tc-antibiotics [17,18] were reported. Among the ^99m^Tc labeled antibiotics, [^99m^Tc]Tc-ciprofloxacin has gained ample attention due to its high target specificity and discrimination potential between bacterial infection and inflammation. It was approved for clinical practice and marketing with the trade name “infecton^®^” [19]. In later reports, bacterial resistance to ciprofloxacin appeared to hamper the enthusiasm [20]. Following the promising infection imaging results of [^99m^Tc]ciprofloxacin, many other fluoroquinolone and cephalosporin antibiotics were labeled with ^99m^Tc, which include [^99m^Tc]Tc-pefloxacin, [^99m^Tc]Tc-ofloxacin, [^99m^Tc]Tc-sparafloxacin, [^99m^Tc]Tc-moxifloxacin, [^99m^Tc]Tc-gemifloxacin, [^99m^Tc]Tc-rufloxacin, [^99m^Tc]Tc-clinafloxacin, [^99m^Tc]Tc-sitafloxacin, [^99m^Tc]Tc-levofloxacin, [^99m^Tc]Tc-ceftriaxone, [^99m^Tc]Tc-cefotaxime, [^99m^Tc]Tc-cefoperazone, and [^99m^Tc]Tc-ertapenem [2,21,22,23,24,25]. The reported results of these radiopharmaceuticals show one or more of the following issues: lipophilicity, non-predicted target specificity, and renal excretion. These issues need to be addressed for its selection to practice clinically. 

Tigecycline (Figure 1), structurally related to minocycline antibiotic, is the first member of the glycylcycline antibiotic class that offers broad-spectrum antibacterial activity and the least susceptibility to develop bacterial resistance. Tigecycline has been reported for the treatment of complicated intra-abdominal infections [26]. Vancomycin-resistant *Enterococci*, methicillin-resistant *S. aureus*, extended-spectrum β-lactamase-producing *Enterobacteriaceae*, and penicillin-resistant *Streptococcus pneumonia* have all been successfully treated with tigecycline [27]. The activity spectrum shows the binding of tigecycline to the bacterial ribosome to inhibit protein synthesis and, consequently, causes the inhibition of bacterial growth [28]. Therefore, tigecycline could be a good choice to develop new ^99m^Tc-labeled radiopharmaceutical for imaging broad-spectrum bacterial infections and to address the issues associated with predecessor radiopharmaceuticals. This study comprises the labeling of tigecycline with ^99m^Tc, quality control analysis, biodistribution in infection-induced rat models, and [^99m^Tc]Tc-tigecycline scintigraphy studies.

## 2. Results

### 2.1. Physical Characteristics 

The reaction mixture of ^99m^Tc labeled tigecycline showed an odorless pale yellow transparent solution. No visible particle debris was seen in the reaction mixture. 

### 2.2. Effect of Quality Control Parameters on Labeling Yield

#### 2.2.1. Effect of pH

The effect of pH on the radiosynthesis of ^99m^Tc-tigecycline was studied at different pH, i.e., 3.5 to 8. The radiochemical yield (RCY) at different reaction pH is shown in Figure 2a. At pH 7.7–7.9, 91% [^99m^Tc]Tc-tigecycline complex, ~7% hydrolyzed activity, and ~2% free ^99m^TcO_4_^−^ was recorded. Other than 7.7–7.9 pH values, the labeling yield remained below 91%.

#### 2.2.2. Effect of Ligand

The effect of the amount of tigecycline on RCY is shown in Figure 2b. The highest RCY, ~91%, was obtained at 0.5 mg/mL ligand concentration. At higher concentrations, a mild discernible change in RCY was noted.

#### 2.2.3. Effect of Reducing Agent 

Figure 2c shows the effect of the stannous chloride dihydrate (SnCl_2_·2H_2_O) reducing agent on radiolabeling yield. The maximum RCY (91%) was obtained by using 100 μg/mL reducing agent concentration.

#### 2.2.4. Effect of Reaction Time 

Figure 2d illustrates the yield of [^99m^Tc]Tc-tigecycline at 5, 15, 25, and 35 min of reaction periods. The maximum yield was achieved after 25 min of reaction period.

### 2.3. Radiolabeling Studies

The percentage of radiochemical ([^99m^Tc]Tc-tigecycline) and radioactive impurities, such as reduced/hydrolyzed ^99m^Tc and free pertechnetate (^99m^TcO_4_^−1^), was assessed using ITLC-SG and ray-HPLC. Figure 3 shows the development of ITLC-SG trip in two different mobile phase systems to evaluate the percent formation of [^99m^Tc]Tc-tigecycline, reduced/hydrolyzed ^99m^Tc, and free ^99m^TcO_4_^−1^. Ray-HPLC analysis, using aqueous ACN doped with 0.1% TFA, was also performed to identify the percentage of RCP. The ray-HPLC radio-chromatogram (Figure 4) shows two well-separated peaks, one at 6.9 min belong to free ^99m^TcO_4_^−1^, and the other at 19.7 min belonging to [^99m^Tc]Tc-tigecycline.

### 2.4. In Vitro Stability of the Complex in Saline

The radiochemical mixture was allowed to incubate at room temperature for 6 h. At predefined time points, an aliquot of 100 µL was taken and analyzed using ITLC-SG analysis to investigate the intact percentage of [^99m^Tc]Tc-tigecycline. The analysis revealed 90% RCP at two hours after the incubation period. However, at successor time points, i.e., at 3, 4, 5, and 6 h, it gradually degraded. The detailed intact RCP at different time points is shown in Figure 5.

### 2.5. In Vitro Bacterial Binding of [^99m^Tc]Tc-Tigecycline and Tigecycline

The bacterial growth inhibition study of [^99m^Tc]Tc-tigecycline and tigecycline was tested against *S. aureus* (Gram-positive) and *E. coli* (Gram-negative) bacterial strains. The results are shown in Figure 6. [^99m^Tc]Tc-tigecycline and tigecycline showed 22.16 ± 1.03 and 12.54 ± 0.82 mm ZOI against *S. aureus* and 21.34 ± 1.61 and 12.67 ± 01.93 mm ZOI against *E. coli*, respectively. The results imply that the radiolabeling does not affect the antibacterial potential of tigecycline.

### 2.6. Partition Coefficient Factor Study

The distribution of [^99m^Tc]Tc-tigecycline between octanol and water solvent (o/w) revealed −1.248 ± 0.01 log P value, which represents the partition coefficient. This study reflected the hydrophilic nature of [^99m^Tc]Tc-tigecycline. The results of triplicate experiments to calculate the log p value are shown in Figure 7.

### 2.7. Biodistribution Study

The biodistribution of [^99m^Tc]Tc-tigecycline was ascertained using healthy and infection-induced rat models. The results of the biodistribution study are presented in Figure 8. The results of these experiments revealed 2.9 ± 0.19 and 2.39 ± 0.31 T/NT values in *S. aureus* and *E. coli* infection-induced rat models, respectively.

### 2.8. Scintigraphy Study of [^99m^Tc]Tc-Tigecycline

Scintigraphy of [^99m^Tc]Tc-tigecycline were performed at 0, 1, 2, and 4 h using an SPECT camera. The time of [^99m^Tc]Tc-tigecycline injection into the animal body was designated as the 0 h time point. The static scintigram of all four time points are shown in Figure 9, which reveals the radiochemical accumulation, washed-out, and excretory path of activity.

## 3. Discussion

Deep-seated bacterial infections are a critical public health problem, and new promising NMI procedures for specifically diagnosing the bacterial infections are needed. The ability to specifically target the bacterial infection even in the presence of inflamed tissues could have a wide range of advantages beyond the diagnosis of bacterial infections. Here, we developed ^99m^Tc-labeled tigecycline to diagnose broad-spectrum bacterial infections. A variety of radioisotopes such as flourin-18, gallium-67, indium-111, etc., are also used for positron emission tomography (PET) and SPECT imaging techniques [18,29,30]; however, due to its most favorable characteristics, ^99m^Tc is preferred in SPECT imaging procedures [31]. Tigecycline is the first member of the glycylcycline antibiotic class. It binds to the bacterial 30S ribosome, blocking the entry of transfer RNA, which results in the inhibition of mitochondrial oxidative phosphorylation and, finally, leads to cell death. For the development of the [^99m^Tc]Tc-tigecycline radiopharmaceutical, the effect of several parameters, including tigecycline quantity, amount of SnCl_2_·2H_2_O, pH, and reaction period, were examined to achieve the highest RCY and least radioactive contaminants. Initially, the hit and trial experiments were conducted to obtain an idea about feasible reaction conditions to reach the optimized values. The reduction of ^99m^Tc from its +7 oxidation state to a lower one is mandatory in the chemistry of ^99m^Tc labeled compounds. A variety of different reducing agents for the reduction of ⁹⁹^m^Tc have been reported in the literature, such as stannous citrate, stannous tartrate, stannous chloride dihydrate, formamidine sulfinic acid, concentrated HCl, sodium borohydride, dithionite, and ferrous sulfate [21]. Due to negligible interference and promising stability at high temperature; SnCl_2_·2H_2_O is preferred for use as a reducing agent in ^99m^Tc-labeling chemistry [17,32]. However, excess amounts of SnCl_2_·2H_2_O may distort the biodistribution profile and limit the diagnostic value of the ^99m^Tc-labeled radiopharmaceuticals [33]. The effect of the SnCl_2_·2H_2_O reducing agent on RCY was studied at 50, 100, 200, 400, 600, 800, 1000, and 1200 μg/mL concentrations. Our investigations revealed that the 100 μg/mL SnCl_2_·2H_2_O concentration is an optimal amount for the maximum reduction of ^99m^Tc to give 91% RCY.

The pH of the reaction mixture is another important reaction parameter in the development of the radiopharmaceutical. It provides the framework to stabilize the radiopharmaceutical and to prevent radiochemical precipitation. A small shift in pH value may alter the interaction between radioisotope and ligand molecules, so maintaining the hydrogen ion concentration is essential to stabilizing the complex. A wide range of reaction pH was investigated by keeping the other parameters constant. At acidic pH, very poor RCY was noted, which gradually increased to 91% at pH 7.7 to 7.9. This reflects that the hydrogen ion concentration does not favor the complexation of ^99m^Tc with tigecycline. The ligand concentration, typically, does not affect RCY if it is stoichiometrically equal to the metal ion. However, by changing the ligand concentration at a constant ^99m^Tc ion concentration, the RCY may affect it, but not drastically. In this study at specific reaction conditions, a 0.5 mg/mL ligand concentration was found to give maximum RCY. Optimum conditions, such as 0.5 mg/mL tigecycline, 100 μg/mL SnCl_2_·2H_2_O, ~185 MBq ^99m^Tc O_4_^−1^, 7.7–7.9 pH, and 25 min reaction period at room temperature (25 ± 2 °C), give 91% RCY. Regarding the coordination chemistry between reduced ^99m^Tc and tigecycline, the amido and hydroxyl groups from two tigecycline molecules may coordinate with ^99m^Tc under optimized reaction conditions. In saline medium, the radiochemical was found intact up to the 2 h time point. Most of the nuclear medicine procedures are completed within a 2 h period.

The tigecycline exhibits broad-spectrum activity against Gram-positive and Gram-negative bacteria. An in vitro antibacterial study showed higher bacterial inhibition potential of [^99m^Tc]Tc-tigecycline as compared to tigecycline. This increased potential of bacterial growth inhibition is either due to the structural modification or ionizing radiation. In both cases, the [^99m^Tc]Tc-tigecycline could be a good choice for the diagnosis of bacterial infection. However, according to the reported studies, the ^99m^Tc labeling with antibiotics does not show a prominent change in antibiotic-bacteria binding efficiency.

The biodistribution of [^99m^Tc]Tc-tigecycline in a rat model revealed intriguing patterns. The quick accumulation of the radiopharmaceutical was noted in the stomach, liver, heart, and kidneys, but other organs did not show the uptake, which may be an indication of the non-toxic nature of newly developed radiopharmaceuticals. The biodistribution pattern at initial time points showed accumulation of [^99m^Tc]Tc-tigecycline, while at later time points, the activity was gradually washed out and accumulated in the bladder after renal filtration. Specifically, [^99m^Tc]Tc-tigecycline showed significant uptake in tissues infected with *S. aureus* (T/NT = 2.9 ± 0.19) and *E. coli* (T/NT = 2.39 ± 0.31), while inflamed tissues showed a 1.29 ± 0.22 T/NT value. Compared to other antibiotics like [^99m^Tc]Tc-gemifloxacin (T/NT = 2.57 ± 0.84) [34], [^99m^Tc]Tc-ceftriaxone (T/NT = 2.24 ± 0.23) [35], [^99m^Tc]Tc-clindamycin (T/NT = 2.37 ± 0.5) [36], and [^99m^Tc]Tc-enrofloxacin (T/NT = 2.84 ± 0.63) [37], [^99m^Tc]Tc-tigecycline demonstrated higher T/NT values in both *S. aureus* and *E. coli*-infected rat models. In this study, the results of biodistribution are in good agreement with the results reported by Ozdemir and colleagues obtained by the labeling of doxycycline hyclate with ^99m^Tc. They reported 94.02 ± 0.41% stability up to 24 h and 2.15 ± 0.39 T/NT ratio in an *E. coli* bacterial infection-induced mice model [38]. Furthermore, the hydrophilic character of the [^99m^Tc]Tc-tigecycline complex demonstrates the reason behind the radiotracer’s quick blood clearance. A scintigraphy study also demonstrated the rapid blood clearance and renal filtration. The scintigraphy study results corroborate the pharmacokinetic and biodistribution findings. Following the intravenous injection of [^99m^Tc]Tc-tigecycline, the scintigraphy study at 0 and 1 h time points reflects radiochemical accumulation in the stomach, heart, liver and kidneys, while at the 4 h time point, approximately 95% of the activity had been filtered through the kidneys and accumulated in the urinary bladder. The radioactivity uptake in the liver and kidneys was likely due to metabolic processes and excretory pathways.

## 4. Materials and Methods

### 4.1. Chemicals and Radioisotope

All the chemicals used in this study were of analytical grade. Hydrochloric acid, acetic acid, sodium chloride, trichloroacetic acid (TCA), acetonitrile, acetone, and SnCl_2_·2H_2_O were acquired from Sigma-Aldrich (Frankfurt, Germany). The local pharmacy provided the tigecycline injection (TYGACIL^®^). Technetium-99m was eluted from a Pakistan ^99^Mo/^99m^Tc generator (PAKGEN) in the form of Na^99m^TcO_4_ at the Punjab Institute of Nuclear Medicine (PINUM), Faisalabad, Pakistan.

### 4.2. Utensil and Equipment

In this study, pyrogen-free glassware and other utensils were used. Agilent Technologies (Frankfurt, Germany) supplied instant ITLC-SG sheets, NaI well-type gamma counter, and 2π-Scanner. A Hitachi L2200 ray-HPLC system (Germany) equipped with a C-18 reverse phase column was used for chromatography analysis.

### 4.3. Bacterial Strains and Animals

Bacterial strains *S. aureus* and *E. coli* were collected from the Department of Biochemistry, Government College University Faisalabad, Pakistan. For conducting biodistribution, albino white rats were acquired, and for scintigraphy, New Zealand white rabbits were acquired from the Department of Physiology, Government College University Faisalabad. The animals were given easy access to food and water under unstressed environmental conditions. The biodistribution and scintigraphy of [^99m^Tc]Tc-tigecycline were performed according to the PINUM and FELASA guidelines and recommendations [39].

### 4.4. ITLC-SG Analysis of the Radiochemical Mixture

The percentage of radiochemical and radioactive impurities such as reduced/hydrolyzed ^99m^Tc and free ^99m^TcO_4_^−1^, was assessed using ITLC-SG and ray-HPLC. ITLC-SG analysis was performed using acetone as a mobile phase to determine the percentage of free ^99m^TcO_4_^−1^. In this run, free ^99m^TcO_4_^−1^ moved with the solvent front, leaving [^99m^Tc]Tc-tigecycline and hydrolyzed ^99m^Tc at the baseline. The aqueous solution of ACN (2:1 *v/v*) was used to record the radioactive counts of reduced/hydrolyzed ^99m^Tc. In this analysis, the [^99m^Tc]Tc tigecycline followed by free ^99m^TcO_4_^−1^ moved with the solvent front, leaving hydrolyzed ^99m^Tc at the baseline. Using the recorded gamma counts of free ^99m^TcO_4_^−1^ and hydrolyzed ^99m^Tc fraction over the ITLC-SG strip and total activity applied on the ITLC-SG strip, the percentages of free ^99m^TcO_4_^−1^ and hydrolyzed ^99m^Tc were calculated. From these two percentages, the percentage of radiochemical was calculated using the following mathematical equation:% Yield of [^99m^Tc]Tc-Tigecycline = 100 – (% Hydrolyzed ^99m^Tc + % Free ^99m^TcO_4_^−^) 

### 4.5. Ray-HPLC Analysis for RCP Analysis

A ray-HPLC system was used to investigate the RCP following the previously reported protocol [2]. In a nutshell, the study was performed using a ray-HPLC system equipped with a C-18 reverse phase column. The ACN (0.1% TFA) solution was used as mobile phase. The separation was carried out using gradient mobile phase system, i.e., the column was primed with 100% CAN, then the sample was eluted using the following gradient system: 0–5 min, 20% ACN; 5–15 min, 40% ACN; 20–25 min, 50% ACN; and, finally, 10 min, 50–100% ACN. A constant flow rate (1 mL/min) for the mobile phase was maintained.

### 4.6. Radiosynthesis of [^99m^Tc]Tc-Tigecycline

The radiolabeling reactions were carried out under a set of different reaction conditions. Serial variation in reaction parameters such as SnCl_2_·2H_2_O concentration, tigecycline concentration, pH, and reaction time at room temperature was investigated. The amount of radioactivity (185 MBq) was chosen to be constant. In all experiments, freshly eluted ^99m^TcO_4_^−1^ was used to label tigecycline. Initially, the hit and trial experiments were conducted to obtain an idea about feasible reaction conditions for further optimization. Finally, the reaction conditions were optimized by conducting the radiolabeling reaction in a serial range such as concentration of tigecycline from 0.25–1.0 mg/mL, SnCl_2_·2H_2_O from 50–1200 μg/mL, pH from 3.5–8, and reaction time 5–35 min. The total volume of the reaction was taken as 1 mL. A glass vial containing all the chemicals was vortexed followed by the addition of a 185 MBq saline solution of ^99m^TcO_4_^−1^. ITLC-SG and ray-HPLC were used to analyze the RCY.

#### Effect of Reaction Parameters

The effect of each parameter; specifically, tigecycline and reducing agent concentration, pH, and reaction time at a particular set of reaction conditions, i.e., by sequential change in the value of one parameter and keeping all other reaction parameter values constant, were recorded. 

### 4.7. In Vitro Stability of [^99m^Tc]Tc-Tigecycline

The saline solution of ^99m^Tc labeled tigecycline was allowed to stand at room temperature for up to 200 min. At predefined time points (starting from the completion of radiolabeling reaction), i.e., 0, 50, 100, and 200 min, an aliquot of 10 μL reaction mixture was drawn and spotted at the baseline of the ITLC-SG strip to evaluate the labeling stability of the complex at each time point.

### 4.8. In Vitro Bacterial Binding Study

The disc diffusion assay was followed to evaluate the binding of [^99m^Tc]Tc-tigecycline and tigecycline with *S. aureus* and *E. coli* bacterial strains [40]. After autoclaving the molten Mueller Hinton agar at 121.7 °C and sterilization, it was transferred into the Petri plate for solidification. Then, in separate plates, the *S. aureus* and *E. coli* bacterial strains were spread over the surface of solid agar using sterile cotton swabs. Following this step, the filter paper discs loaded with the sample at the appropriate concentration were placed on the agar surface. The plates were then incubated at 37 °C for 24 h. At the end of the incubation period, the diameter of zone of inhibition (ZOI) was measured in millimeters (mm).

### 4.9. Partition Coefficient (Log P) Measurements

The partition coefficient of the radiochemical was determined as follows: a solution of 1 mL n-octanol and 900 μL water was mixed with 100 μL of the radiopharmaceutical, and then the mixture was blended well using a vortex mixer. The mixture was centrifuged at 3000 rpm for 5 min to separate the two layers. A 100 μL aliquot was drawn from each layer, and a NaI-well type gamma ray scintillation counter was used to count the radioactivity in two fractions independently. The experiment was conducted three times (*n* = 3, ±S.D.), and the expression log P(o/w) was used to express the final partition coefficient.

### 4.10. Biodistribution [^99m^Tc]Tc-Tigecycline in Rats

The albino white rats were used to conduct [^99m^Tc]Tc-tigecycline biodistribution in bacterial infection and inflammation-induced rat models following the protocol reported in the literature [41]. The infection and inflammation was induced in right thigh muscles using bacterial strains and turpentine oil 48 h before the biodistribution study. The albino white rats were split into three groups (*n* = 3 per group): rats infected with *S. aureus* (group I), rats infected with *E. coli* (group II), and rats with induced inflammation (group III). Once the animals showed visible redness and swelling after 48 h of infection and inflammation induction, an aliquot of 100 μL of [^99m^Tc]Tc-tigecycline (37 MBq) was administrated intravenously into the rat’s tail vein. The animals were then anesthetized using chloroform at 30 min, 2 h, 4 h, and 6 h time points and sacrificed. The key body organs, such as small and large intestine, stomach, spleen, liver, lungs, heart, kidney, reproductive organ, and infected (target)/inflamed (non-target) and normal thigh muscles, were then removed surgically with a professional surgical blade/seizer, washed with distilled water to remove surface-bound activity, weighed, and, finally, stored in γ-counting tubes. The radioactivity accumulated in each organ was recorded using a well-type NaI γ-counter. The observed counts were calibrated using the corresponding decay of the ^99m^Tc isotope and represented as a percentage of the administered dose per gram (%ID/g) organ.

### 4.11. [^99m^Tc]Tc-Tigecycline Scintigraphy Study

Normal rabbit models were utilized for the [^99m^Tc]Tc-tigecycline scintigraphy studies. The rabbits were anesthetized by intramuscular injection of diazepam with a dose limit of 20 mg/kg, followed by the injection of 300 μL [^99m^Tc]Tc-tigecycline (385 MBq) through inferior ear vein under pyrogen-free conditions. Scintigraphy was performed using a double-headed Siemens gamma camera (Germany) attached to the Xeleris Workstation, version 4. Each animal was placed on a flat, hard surface with both fore and hind legs spread out (fixed with surgical tape) to record static and dynamic SPECT imaging. The whole body scan was performed at 0, 1, 2, and 4 h post-injection time points.

### 4.12. Statistical Analysis

All the results obtained from different experiments were analyzed statistically using 1-way analysis of variance using Tukey test and expressed as the mean of three experimental (*n* = 3) values along with standard error of mean (mean ± S.D.). *p* value < 0.05 was representing the significant difference between groups.

## 5. Conclusions

In this prospective study, [^99m^Tc]Tc-tigecycline was prepared with 91% RCY. The radiopharmaceutical, in addition, exhibited good post labeling stability, which ensured safe administration and minimum radiotoxicity. Biodistribution data reflected minimal accumulation in normal body tissues and complete washout within 4 h of administration. The T/NT values associated with *S. aureus* infection (2.9 ± 0.19), *E. coli* infection (2.39 ± 0.31), and inflamed tissues (1.29 ± 0.22) show the ability of [^99m^Tc]Tc-tigecycline to image bacterial infection specificity. The dynamic SPECT scintigraphy data of [^99m^Tc]Tc-tigecycline reflects minimal renal retention and normal perfusion rate. In conclusion, the newly developed [^99m^Tc]Tc-tigecycline radiopharmaceutical could be a fruitful addition to nuclear medicine procedures to diagnose deep-seated bacterial infections.

## Figures and Tables

**Figure 1 pharmaceuticals-17-01283-f001:**
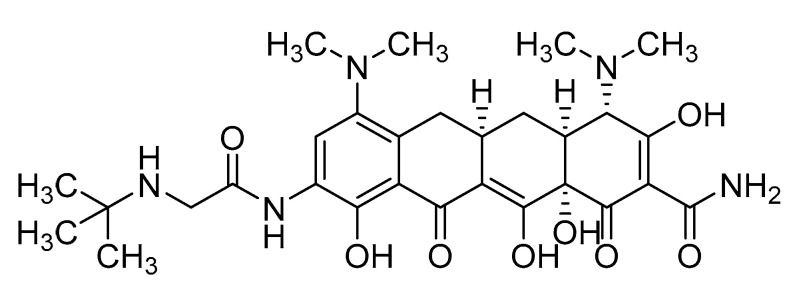
Chemical structure of tigecycline antibiotic.

**Figure 2 pharmaceuticals-17-01283-f002:**
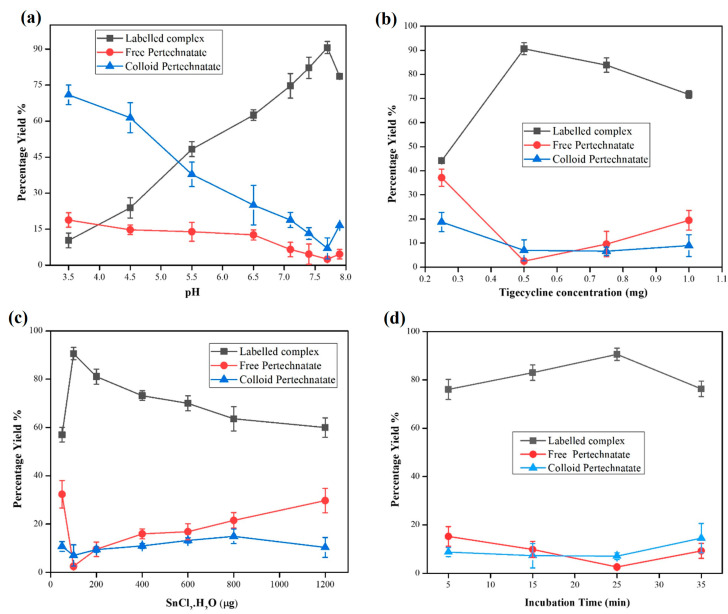
Study of the effect of pH (**a**), ligand concentration (**b**), reducing agent (**c**), and reaction period (**d**). All values are represented as mean (*n* = 3) ± S.D.

**Figure 3 pharmaceuticals-17-01283-f003:**
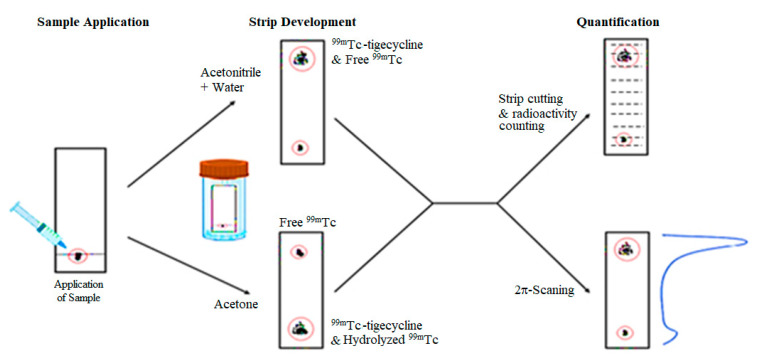
ITLC-SG analysis description for the determination of [^99m^Tc]Tc-tigecycline, free ^99m^TcO_4_^−^, and hydrolyzed ^99m^Tc.

**Figure 4 pharmaceuticals-17-01283-f004:**
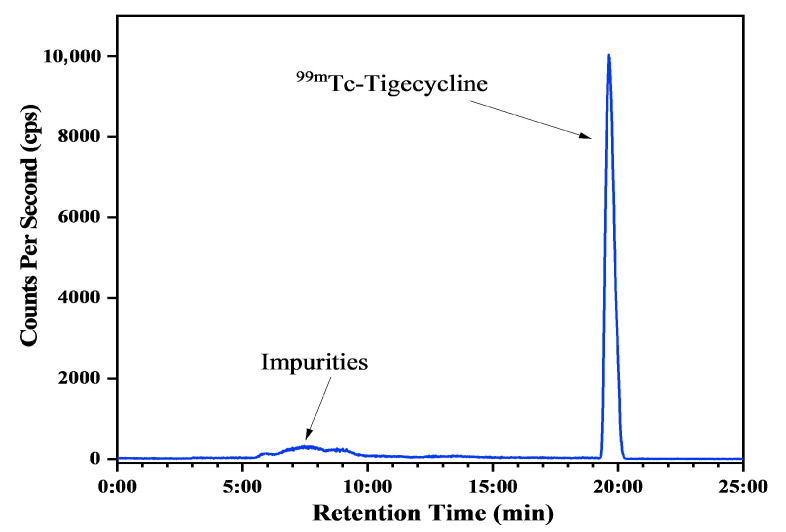
Ray-HPLC radio-chromatogram showing the peaks of [^99m^Tc]Tc-tigecycline and impurities.

**Figure 5 pharmaceuticals-17-01283-f005:**
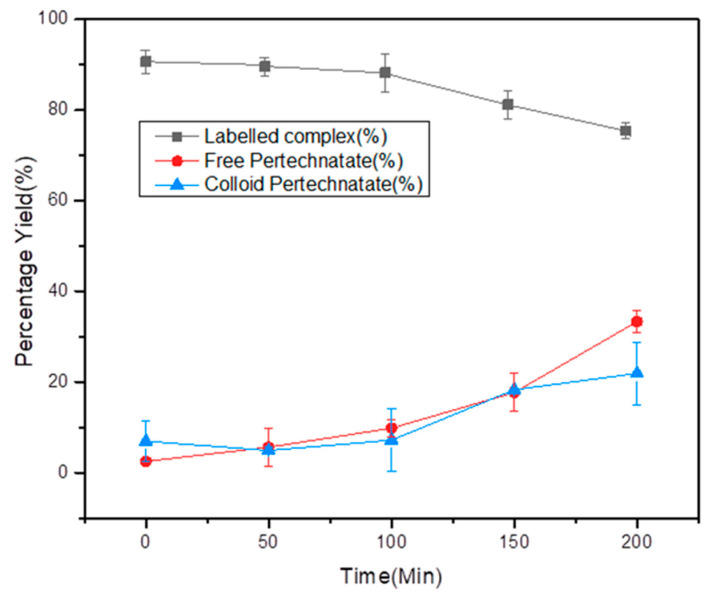
In vitro stability of the radiochemical in saline. All values are represented as mean (*n* = 3) ± S.D.

**Figure 6 pharmaceuticals-17-01283-f006:**
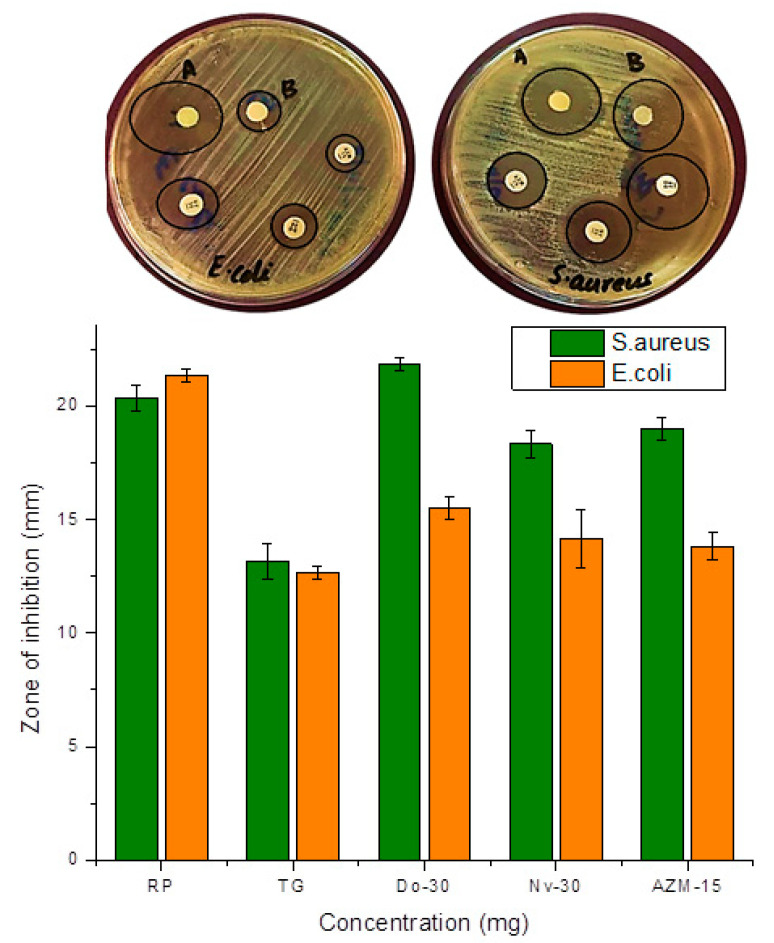
Antibacterial activity of [^99m^Tc]Tc-tigecycline (A) and untagged tigecycline (B) against *S. aureus* and *E. coli* bacterial strains. All values are represented as mean (*n* = 3) ± S.D.

**Figure 7 pharmaceuticals-17-01283-f007:**
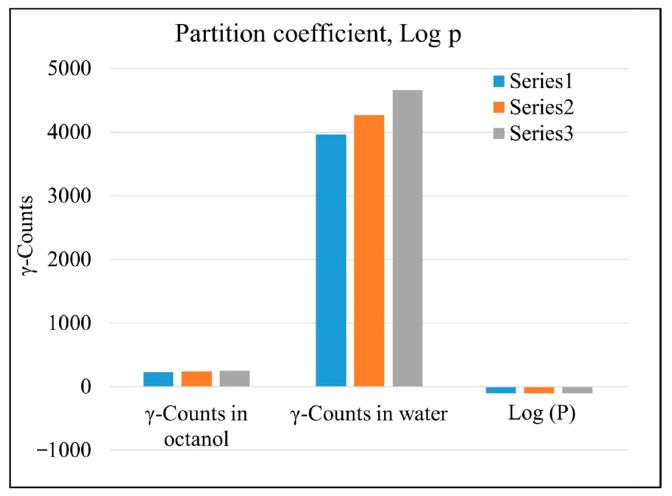
Distribution of ^99m^Tc labeled tigecycline in octanol and water.

**Figure 8 pharmaceuticals-17-01283-f008:**
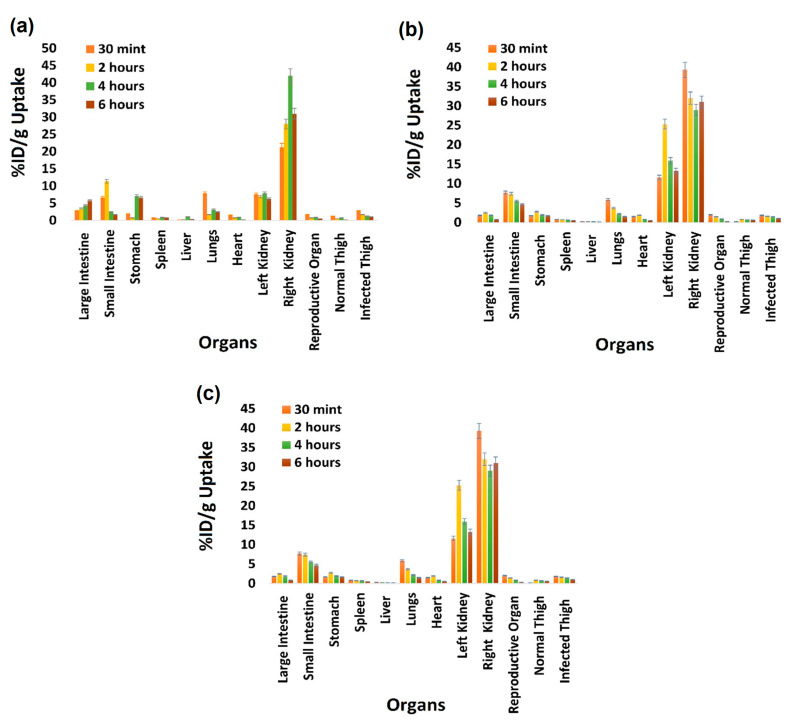
Biodistribution in rats infected with (**a**) *S. aureus* bacterial strain, (**b**) with *E. coli* bacterial strain, and (**c**) turpentine oil (inflammation). All values are represented as mean (*n* = 3) ± S.D.

**Figure 9 pharmaceuticals-17-01283-f009:**
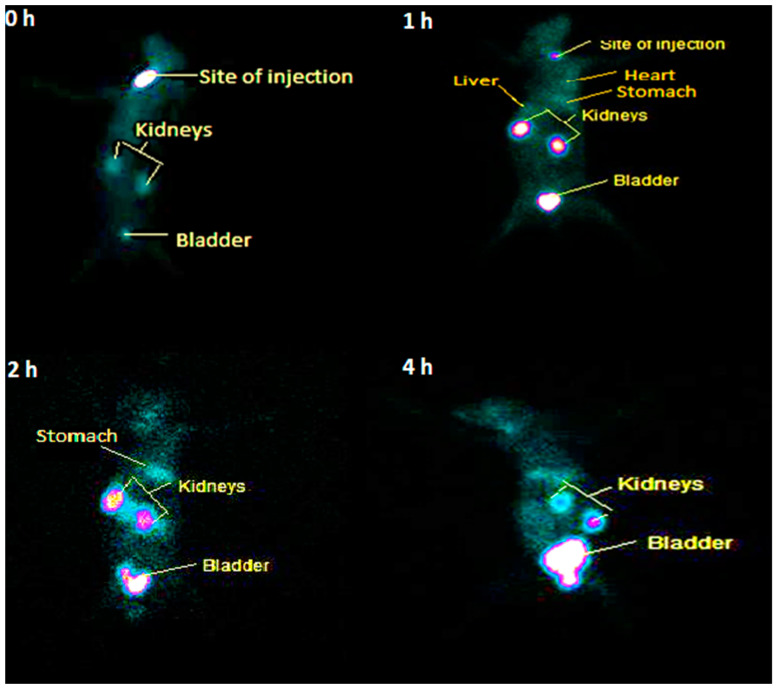
Scintigraphy scan of [^99m^Tc]Tc-tigecycline in normal rabbit at 0 h, 1 h, 2 h, and 4 h post injection period.

## Data Availability

Data is contained within the article.

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
