# Peer review of "Radiosynthesis and Preclinical Evaluation of [99mTc]Tc-Tigecycline Radiopharmaceutical to Diagnose Bacterial Infections"

_pharmaceuticals, 2024, doi:10.3390/ph17101283_

Round 1

Reviewer 1 Report

Comments and Suggestions for Authors

It is a detailed radiopharmaceutical development study including in vivo data that is very comprehensive, I think that if the authors make a reorganization within the scope of the following suggestions, it will contribute to the enrichment of the study.

1.      I suggest that you make corrections to the English spelling throughout the text,

2.      Please, 99mTcO4 -1, update the spelling of the term to superscript -2,

3.      Where in the animal was the infection model established?

4.      I suggest discussing whether the focus of infection visible in the scintigrams is the thyroid tissue,

5.      expanding the discussion by referencing experimental data and current literature,

6.      You can add information about the elimination pathway in the discussion section,

Comments on the Quality of English Language

It is a detailed radiopharmaceutical development study including in vivo data that is very comprehensive, I think that if the authors make a reorganization within the scope of the following suggestions, it will contribute to the enrichment of the study.

1.      I suggest that you make corrections to the English spelling throughout the text,

2.      Please, 99mTcO4 -1, update the spelling of the term to superscript -2,

3.      Where in the animal was the infection model established?

4.      I suggest discussing whether the focus of infection visible in the scintigrams is the thyroid tissue,

5.      expanding the discussion by referencing experimental data and current literature,

6.      You can add information about the elimination pathway in the discussion section,

Author Response

Response to Reviewer 1 Comments

Our sincere thanks to the Reviewers for their comments, which have helped improve the quality of this manuscript. Below are the detailed responses to reviewer’s comments and suggestions. The corresponding revisions/corrections highlighted/in red font in the re-submitted files

2. Point-by-point response to Comments and Suggestions for Authors

It is a detailed radiopharmaceutical development study including in vivo data that is very comprehensive, I think that if the authors make a reorganization within the scope of the following suggestions, it will contribute to the enrichment of the study.

Comments 1:

I suggest that you make corrections to the English spelling throughout the text

Response 1:  

Thank you for pointing this out. We agree with this comment. The entire manuscript was reviewed for grammatical and technical accuracy. Ambiguous, or unclear phrases were reformulated to be more concise and aligned with the objectivity pertinent to a scientific manuscript. Grammatical errors were thoroughly checked and corrected.

Comments 2:

Please, 99mTcO4 -1, update the spelling of the term to superscript -2

Response 2:

The spelling was updated and also fixed the superscript mistake.

Comments 3:

Where in the animal was the infection model established?

Response 3:

The infection was induced in right thigh muscles of Albino white rats in three groups. The detailed animal model protocol is mentioned under heading 4.10 Biodistribution (L354) 

Comments 4:

I suggest discussing whether the focus of infection visible in the scintigrams is the thyroid tissue,

Response 4:

Thanks, the scintigraphy of [99mTc]Tc-tigecycline was performed in normal Rabbits therefore, no sign of infection is possible in thyroid tissues.

Comments 5:

expanding the discussion by referencing experimental data and current literature.

Response 5:

Thank you for suggestion to improve the discussion section. We have added more reported data with reference citation to improve this section.

Comments 6:

You can add information about the elimination pathway in the discussion section.

Response 6: In the revised version L259-L261 describe excretory pathway.

Reviewer 2 Report

Comments and Suggestions for Authors

Dear autors

I find you article relevant because their for sure is a need for developing new tracers to diagnosing bacterial infection as well as sterile inflammation. I think your work could be published after a major revision.

Frist of all I think you have to rearrange the article quite a lot, results should be in the results part, discussion in the discussion section and so on

I think most of what you have written in the results section aren’t results, so should either be deleted or moved to for example the discussion part.

I think the results are:

-        The optimized synthesis (how you did the optimization should go into the discussion manly and how you did the experiment in to materials and methods). In the discussion you discuss the influence of pH, ligand concentration, reducing agent and reaction time)

-        The result of the labeling should go in the results section, but the test set up and the QC set up should go into the materials and method section

-        For the biodistribution study, I believe how things are done should be moved to the materials and method section.

Under materials and methods. Under 4.1. Chemicals, you should only have chemicals, so you (maybe) need a heading for utensil and equipment and one for animals.

You need more into on the animals, what kind of rabbit, how long before the experiments where they obtained, how were they keep, access to food and water? It is unclear to me which experiments you used rats and which you used rabbits for

Consensus nomenclature rules for radiopharmaceutical chemistry nomecalature should be followed so it should be [99mTc]Tc-tigecycline and not just 99mTc-tigecycline (throughout the paper)

L62-L67. It is unclear to me why you list the radiopharmaceuticals you do and why not some other ones maybe just refer to a few reason reviews, for eksample

1) Radiopharmaceuticals for PET and SPECT Imaging: A Literature Review over the Last Decade.

Crișan G, Moldovean-Cioroianu NS, Timaru DG, Andrieș G, Căinap C, Chiș V.

Int J Mol Sci. 2022 Apr 30;23(9):5023. doi: 10.3390/ijms23095023.

PMID: 35563414 Free PMC article. Review.

2) Radiotracers for Bone Marrow Infection Imaging.

Jødal L, Afzelius P, Alstrup AKO, Jensen SB.

Molecules. 2021 May 25;26(11):3159. doi: 10.3390/molecules26113159.

PMID: 34070537 Free PMC article. Review.

3) Radiotracer Development for Bacterial Imaging.

Mota F, Ordonez AA, Firth G, Ruiz-Bedoya CA, Ma MT, Jain SK.

J Med Chem. 2020 Mar 12;63(5):1964-1977. doi: 10.1021/acs.jmedchem.9b01623. Epub 2020 Feb 21.

PMID: 32048838 Free PMC article. Review.

In line 23 you say, “to treat infectious diseases”, please give examples/references for that statment

Please go though the abstract and make sure it that it agrees with the article and methods are described in the method section, etc

Is it fair to say that you have looked at a dread spectrum of bacteria when you have only looked at two different ones?

In L258, you compare [99mTc]Tc-tigecycline to other antibiotics like  [99mTc]Tc-gemifloxacin, [99mTc]Tc-cerftriaxone, [99mTc]Tc-cidamycin  and  [99mTc]Tc-enrofloxacin, but you don’t tell us if any of them is in clinical use anywhere, please to so.

L183 “inthigh” I assume “in thigh”?

L 396 “era” should it be area?

L417,  Pakistan, I assume

Author Response

Response to Reviewer 2 Comments

1. Summary

Our sincere thanks to the Reviewers for their comments, which have helped improve the quality of this manuscript. Below are the detailed responses to reviewer’s comments and suggestions. The corresponding revisions/corrections highlighted/in red font in the re-submitted files

2. Point-by-point response to Comments and Suggestions for Authors

I find you article relevant because their for sure is a need for developing new tracers to diagnosing bacterial infection as well as sterile inflammation. I think your work could be published after a major revision.

Comments 1:

Frist of all I think you have to rearrange the article quite a lot, results should be in the results part, discussion in the discussion section and so on

Response 1:  

Thank you for pointing this out. We agree with this comment. Therefore, we have rearranged the text according to the comment.

Comments 2:

I think most of what you have written in the results section aren’t results, so should either be deleted or moved to for example the discussion part.

Response 2:

Thank you for pointing this out. We agree with this comment. Therefore, we have put the right part in right section. Most of the text part that was mentioned in Results section (section 2.1 to 2.8), have been shifted to discussion section and some part to Materials and Method section.

Comments 3:

I think the results are:

- The optimized synthesis (how you did the optimization should go into the discussion manly and how you did the experiment in to materials and methods). In the discussion you discuss the influence of pH, ligand concentration, reducing agent and reaction time)

Response 3:

Thank you for pointing this out. In Results section we only have mentioned results and irrelevant text either we have shifted to relevant section or deleted. Further, in discussion section we have discussed the influence of pH, ligand concentration, reducing agent and reaction time to support the results.

Comments 4:

- The result of the labeling should go in the results section, but the test set up and the QC set up should go into the materials and method section

Response 4: The result of the labeling have been shifted to results section, and the test and QC set up shifted to the materials and method section

Comments 5:

- For the biodistribution study, I believe how things are done should be moved to the materials and method section.

Response 5:

Thank you for pointing this out. The experimental protocol shifted to the materials and method section from Results section.

Comments 6:

Under materials and methods. Under 4.1. Chemicals, you should only have chemicals, so you (maybe) need a heading for utensil and equipment and one for animals.

Response 6: The section 4.1 divided into three section i.e. 4.1 Chemicals and radioisotope; 4.2 Utensils and equipment; and 4.3 Bacterial strains and animals

Comments 7:

You need more into on the animals, what kind of rabbit, how long before the experiments where they obtained, how were they keep, access to food and water? It is unclear to me which experiments you used rats and which you used rabbits for

Response 7:

Thank you for pointing this out. Added all the information under 4.3 heading (L277 – L280). 

Comments 8: Consensus nomenclature rules for radiopharmaceutical chemistry nomecalature should be followed so it should be and not just  (throughout the paper)

Response 8: Thanks. Replaced 99mTc-tigecycline with [99mTc]Tc-tigecycline throughout the manuscript.

Comments 9:

L62-L67. It is unclear to me why you list the radiopharmaceuticals you do and why not some other ones maybe just refer to a few reason reviews, for example

1) Radiopharmaceuticals for PET and SPECT Imaging: A Literature Review over the Last Decade. Crișan G, Moldovean-Cioroianu NS, Timaru DG, Andrieș G, Căinap C, Chiș V. Int J Mol Sci. 2022 Apr 30;23(9):5023. doi: 10.3390/ijms23095023. PMID: 35563414 Free PMC article. Review.

2) Radiotracers for Bone Marrow Infection Imaging. Jødal L, Afzelius P, Alstrup AKO, Jensen SB. Molecules. 2021 May 25;26(11):3159. doi: 10.3390/molecules26113159. PMID: 34070537 Free PMC article. Review.

3) Radiotracer Development for Bacterial Imaging. Mota F, Ordonez AA, Firth G, Ruiz-Bedoya CA, Ma MT, Jain SK. J Med Chem. 2020 Mar 12;63(5):1964-1977. doi:10.1021/acs.jmedchem.9b01623. Epub 2020 Feb 21. PMID: 32048838 Free PMC article. Review.

Response 9: Thanks for providing very important review article for making a part in introduction and discussion section. These three references and few others we have cited in in discussion section (Reference 29, 30 & 31). The work cited in L62-L68 belong to other research group.

Comments 10: In line 23 you say, “to treat infectious diseases”, please give examples/references for that statment

Response 10: Thank you for pointing this out. We have edited it as “Nuclear Medicine Imaging (NMI) is known for its promising potential to diagnose deep seated bacterial infections”. 

Comments 11: Please go though the abstract and make sure it that it agrees with the article and methods are described in the method section, etc

Response 11:

Thank you for pointing this out. The mistake have been fixed (L31-L38)

Comments 11: Is it fair to say that you have looked at a dread spectrum of bacteria when you have only looked at two different ones?

Response 12: The selection of these two bacterial strains was according the designed protocol by PINUM Hospital, due to its MDR nature.

Comments 11: In L258, you compare [99mTc]Tc-tigecycline to other antibiotics like  [99mTc]Tc-gemifloxacin, [99mTc]Tc-cerftriaxone, [99mTc]Tc-cidamycin  and  [99mTc]Tc-enrofloxacin, but you don’t tell us if any of them is in clinical use anywhere, please to so.

Response 12: Most probably some of these are in pre-clinical phases e.g. we are studying few such radiotracers clinically.   

Comments 11:

L183 “inthigh” I assume “in thigh”?

L 396 “era” should it be area?

L417,  Pakistan, I assume

Response 12:

Thank you for pointing this out. Corrected all these mistakes. 

Round 2

Reviewer 2 Report

Comments and Suggestions for Authors

Thank you for rearranging the contents of the paper. To me it is almost like reading a different article

A couple og very minor things:

-          L2, L39, 134, 157and L365:  99mTc-tigecycline should be [99mTc]Tc-tigecycline

-           L88 I would say: “Therefore, tigecycline could be a good choice to develop..” or “could potential be”

 -          I would move figure 2 into the the discussion section

 -          L193, please add an argument for why S. aureus + E. coli bacterial is a broad bacterial spectrum

-          L206 Stannous chloride dihydrate instead of SnCl2, 2H2O (either you use names or you use molecular formula, and all the other are names)

-          L214 line breaks between ” RCY.” And “pH”

-          L274 “Punjab Institute of Nuclear Medicine; India”

I think you have done a good job of producing a very interesting article and I’m happy to recommend your article for publication.

Author Response

Response to Reviewer 2 Comments

1. Summary

Our sincere thanks to the Reviewers for the final comments, which have helped to improve the quality of this manuscript. Below are the detailed responses to reviewer’s comments and suggestions. The corresponding revisions/corrections highlighted/in red font in the re-submitted files

2. Point-by-point response to Comments and Suggestions for Authors

Thank you for rearranging the contents of the paper. To me it is almost like reading a different article

A couple of very minor things:

Comments 1:

L2, L39, 134, 157and L365:  99mTc-tigecycline should be [99mTc]Tc-tigecycline

Response 1:  

Thank you for pointing these out, which we have left in revised version. Corrected all.

Comments 2:

L88 I would say: “Therefore, tigecycline could be a good choice to develop..” or “could potential be”

-   I would move figure 2 into the discussion section

Response 2:

Thank you for advising correction. Would have been replaced with could. However, to mention the QC results, we retained the Figure 2 at original place.

Comments 3:

L193, please add an argument for why S. aureus + E. coli bacterial is a broad bacterial spectrum

Response 3:

The L193 is “Here we developed 99mTc-labeled tigecycline to diagnose broad-spectrum bacterial infections” In manuscript.   

Comments 4:

L206 Stannous chloride dihydrate instead of SnCl2, 2H2O (either you use names or you use molecular formula, and all the other are names)

Response 4:

Thanks, the formula was replaced with name

Comments 5:

L214 line breaks between ” RCY.” And “pH”

Response 5:

Break up done between ACY and pH lines.

Comments 6:

L274 “Punjab Institute of Nuclear Medicine; India”

I think you have done a good job of producing a very interesting article and I’m happy to recommend your article for publication.

Response 6:

Country name added. Thanks for kind comments.
